



# Contribution of future wide swath altimetry missions to ocean analysis and forecasting

Antonio Bonaduce [1], Mounir Benkiran [1], Elisabeth Remy [1], Pierre Yves Le Traon [1,2], and Gilles Garric [1]

[1]Mercator Ocean, Toulouse, France
[2]Ifremer, Plouzané, France

*Correspondence to:* A. Bonaduce (antonio.bonaduce@mercator-ocean.fr)

**Abstract.** The impact of forthcoming wide-swath altimetry missions on the ocean analysis and forecasting system was investigated by means of OSSEs (Observing System Simulation Experiments) performed with a regional data assimilation system, implemented in the Iberian-Biscay-Ireland (IBI) region, at 1/12° resolution using simulated observations derived from a fully eddy-resolving free simulations at 1/36° resolution over the same region. The objective was to asses the contribution of dif-

5 ferent satellite constellations to constrain the ocean analyses and forecasts, considering both along-track altimeters and future wide-swath missions, and as consequence the capability of the data assimilation techniques used in Mercator Ocean operational system to effectively combine the different kind of measurements. This was carried out as part of a European Space Agency (ESA) study on the potential role of wide-swath altimetry for the evolution of the European Union Copernicus programme. The impact of future wide-swath altimetry data is clearly evident investigating the reliability of sea-level in the OSSEs. The

10 most significant results were obtained looking at the sensitivity of the system to wide-swath instrumental error: considering a constellation of three nadir and two "accurate" (small instrumental error) wide-swath altimeters, the error in the ocean analysis was reduced up to the 50 %, with respect to conventional altimeters. Investigating the impact of the repetitivity of the future measurements, the results showed that two wide-swath missions had a major impact on the the sea-level forecasting increasing the accuracy over the entire time-window of the 5-day forecasts, with respect to a single wide-swath instrument. A spectral

analysis underlined that the contributions of wide-swath altimetry data observed in the ocean analyses and forecasts statistics were mainly due to resolve more accurately (up to > 25 %), with respect to along-track data, the ocean variability at spatial scales smaller than 100 km. Considering the ocean currents, the results confirmed that the information provided by wide-swath measurements at the surface is propagated also in the vertical and has a considerable impact (30 %) on the ocean currents (up to 300 metres), with respect to the present constellation of altimeters. The ocean analysis and forecasting systems used here

are currently adopted by Copernicus Marine Environment and Monitoring Service (CMEMS) to provide operational services and ocean re-analysis. The results obtained in the OSSEs considering along-track altimeters were consistent with those derived with real data (observing system experiments, OSEs). OSSEs also allow to evaluate the potential of new observing systems and in this study the results showed that future constellations of altimeters will have a major impact to constrain the CMEMS ocean analysis and forecasting systems and their applications.

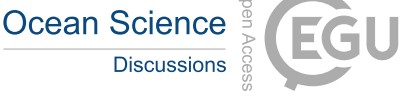

*Copyright statement.* TEXT

# 1 Introduction

Satellite altimetry measurements, based on nadir radar altimeters, have been a fundamental contribution to the understanding of the ocean circulation (Fu and Chelton, 2001; Le Traon and Morrow, 2001), during the past two decades. The continuous
improvement of the physics in the Ocean General Circulation Models (OGCMs) and of data assimilation schemes (DAS) (Bell et al., 2015), nowadays allow exploiting these unique source of information within global and regional ocean monitoring and forecasting systems (Le Traon et al., 2017b). The focus of the present study is to investigate the impact of wide-swath altimetry on a high resolution regional ocean analysis and forecasting system. This is carried out as part of a European Space Agency (ESA) study on the potential role of wide-swath altimetry for the evolution of the European Union Copernicus programme
(space component).

While along-track measurements can observe wavelengths up to 50-70 km (Dufau et al., 2016), the representation of the ocean mesoscale dynamics is strongly limited by the spatial (distance between neighboring tracks) and temporal (repeat period) sampling of a given altimeter mission. The use of multiple altimeters is needed to constrain the mesoscale circulation (Le Traon et al., 2015) and to provide global maps of the mesoscale variability of the ocean (Le Traon and Dibarboure, 1999;
Morrow and Le Traon, 2012). In the literature, the works that focus on the capability of altimeter constellations to resolve the ocean dynamics at the mesoscale resolution (e.g. Dufau et al., 2016), highlighted that at least three altimeters are required to reconstruct sea-level variations (Pascual et al., 2006; Dibarboure et al., 2011) and that the merging of multiple altimeter missions cannot resolve wavelengths smaller than 150 – 200 km (Ducet et al., 2000; Le Traon, 2013). The shortcomings of the conventional altimetry could be addressed by wide-swath measurements of the sea-surface height (SSH) planned for the future
space missions (e.g. the Surface Ocean and Water Topography, SWOT, Mission; Fu et al., 2009; Durand et al., 2010), extending the capability of existing nadir altimeters to two-dimensional mapping and sampling the ocean surface at unprecedented spatial resolution resolution, up to wavelengths of 20 km (Fu and Ferrari, 2008).

The forthcoming altimeter missions based on radar interferometry to obtain wide-swath measurement of sea-surface elevation (Fu et al., 2009), represent the next generation satellite altimetry measurements with a high potential for ocean analysis
and forecasting. The NASA/CNES SWOT mission will be the first swath altimetry mission to be launched in 2021.

Wide-swath altimetry concept is expected to represent an essential contribution to operational oceanography (Bell et al., 2015; Le Traon et al., 2017b). On the other hand, the temporal resolution of wide-swath data, considering SWOT-like orbit parameters (repeat cycle of 21 days with a 10 days subcycle), could be not suitable to resolve the evolution of mesoscale eddies (Ubelmann et al., 2015). A major challenge will be to combine wide-swath and the conventional along-track altimeters (Pujol
et al., 2012) data with high resolution OGCMs to allow a dynamical interpolation of wide-swath data and a detailed description and forecast of the ocean state at high resolution. Approaching the challenge of using wide-swath altimetry data to reconstruct oceanic fields, Gaultier et al. (2016b) underline the need of testing their effective impact on the ocean analyses and forecasts



performing observing system simulation experiments (OSSEs). This work represent a first effort to investigate the impact of forthcoming wide-swath altimetry data in an ocean analysis and forecasting system by means of OSSEs.

Observing sytem experiments (OSEs; e.g. Oke and Schiller, 2007) and OSSEs (e.g. Halliwell et al., 2017) are rigorous methods to provide demonstrations of the impacts of observations (Schiller et al., 2015) on global (Oke et al., 2015a) and regional

(Oke et al., 2015b) ocean forecasting systems, as underlined by the GODAE OceanView International Programme (Bell et al., 2015). OSEs analyse the impact of real observations through data denial experiments, where the impact is determined by the increase in analysis and forecast errors due to neglecting a given observing system (Atlas et al., 2015). OSSEs extend this procedure to the evaluation of new deployment strategies and sampling characteristics for existing systems, and to the design of new observing systems. Observations are, in the case of OSSE, simulated to mimic the sample and error specification of the

future network design and then assimilated.

As described in the literature (e.g. Errico et al., 2013), OSSEs typically use two different OGCMs or two different OGCM configurations. Halliwell et al. (2014) proposed a "fraternal twin" approach, where the same OGCM was used for both the observation simulation and the assimilative model, to evaluate the impact of Earth observations (EO) in the ocean. This approach was adopted also in this study, using two different configurations of the Nucleus for European Modelling of the Ocean (NEMO

; Madec, 2016). A first configuration is used to perform a "nature" run (hereafter NR) to represent the "true" ocean. Synthetic observations are obtained sampling the NR in order to mimic either an existing or future observing system. The synthetic observations are then ingested, through data assimilation techniques, in numerical simulations performed with a second OGCM configuration, to obtain a representation of the state of the ocean constrained by the observing system considered (assimilated run). The impact of the simulated observation system is quantified comparing the assimilated run (AR) against the NR and the

different performances among the OSSEs can be evaluated by the reduction (increase) of ocean analysis and forecast errors due to considering (neglecting) the new observing systems designed.

OSSEs are complementary to OSEs and the results for existing observing systems must be consistent with those derived from OSSEs. In particular, the growth (reduction) of the error among the NR and the assimilated run in the OSSEs should have an order of magnitude comparable to the one obtained in the OSEs comparing a realistic ocean analysis and forecasting

system, which consider real observations, with the real ocean. The calibration of OSSEs with respect to OSEs is an important element to obtain robust results from OSSEs (Halliwell et al., 2014, 2017; Kourafalou et al., 2016). In this sense, the choice of the NR, assimilated run (AR), data assimilation scheme (DAS) and the errors to be considered for the synthetic observations have to be carefully analysed to avoid inconsistent departures of forecast and analysis errors in the OSSEs. In this work we investigated the potential impact of future constellations of satellite altimeters, based on nadir and wide-swath missions, using a

regional ocean analysis and forecasting system implemented in the Iberian-Biscay-Ireland (IBI) region at the spatial resolution of 1/12°. The system was validated against in situ and satellite observations and provide operational services and ocean reanalysis (Sotillo et al., 2015) within the framework of the Copernicus Marine Enviroment and Monitoring Service (CMEMS). The main objective of this study was to quantify the impact of assimilating wide-swath altimetry data on the errors in the ocean analyses and forecasts.





OSSEs are also important tools for testing the capability of the DAS to effectively merge different types of observations with models to produce improved ocean analyses and forecasts. Wide-swath measurement noise, due to radar interferometer instrument, and its cross-track variability within the swath, will also be a complex issue which must be taken into account (Hénaff et al., 2008) to ensure an effective use of the data. In this study a particular attention was given to the sensitivity of the

5 ocean analysis and forecasting system to the instrumental error of wide-swath altimetry measurements. The aim was to test the capability of the Mercator Ocean DAS to use and merge nadir and wide-swath altimeters, which to the best of our knowledge has never been investigated using a regional ocean analysis and forecasting systems.

The paper is organized as follow. Section 2 describes the ocean modelling and data assimilation components of the OSSEs, as well as the synthetic observations considered in the different experiments. The experimental set-up designed to assess the

10 impact of wide-swath altimetry data is detailed in Section 3. The impact of wide-swath data at the surface is investigated in Section 4. The contribution of the new observing systems to the representation of the ocean variability at different spatial scales is evaluated in Section 5. Section 6 shows the impact of wide-swath altimetry data on the ocean circulation, both at the surface and in the water column. All the results are summarized in Section 7 and some conclusions are drawn.

## 2  OSSE approach

ESA is conducting a study to assess the feasibility and potential of wide-swath altimetry for the EU Copernicus programme. The main objective is to provide a much improved operational monitoring of the ocean mesoscale variability for the Copernicus Marine Service (e.g. Le Traon et al., 2017a). Different wide swath altimeter concepts are analyzed by Thales Alenia Space (TAS) as part of this ESA study. Compared to the SWOT mission that is focused on submesoscale variability, these European wide-swath altimetry concepts have less stringent noise measurement requirements. Their potential for ocean analysis and

forecasting are analyzed here by means of OSSEs.

In this Section we describe the OSSEs components, represented by the OGCM configurations used to obtain the synthetic observations and to perform data assimilation experiments, the DAS adopted to consider the new observing systems in the ocean analysis, the simulated ocean observations and their errors.

### 2.1  OGCM configurations

In this study, both NR and the AR rely on the last version of the NEMO OGCM (NEMO v3.6; Madec, 2016). Following a "fraternal twin" aproach (Halliwell et al., 2014), even though the same OGCM type is used , the NR and the AR are configured differently so that the errors (differences between NR and AR) are similar to the one found between state-of-the-art ocean models (e.g. Maraldi et al., 2013; Sotillo et al., 2015) and the true ocean.

The NR is a free running simulation of the NEMO OGCM, implemented in the IBI region at an eddy resolving spatial

resolution and using an explicit free surface formulation (Madec, 2016; Oddo et al., 2014). The primitive equations are discretized on an horizontal curvilinear grid which is a refined subset at 1/36° (2-3 km) of the so-called "ORCA" tripolar grid, commonly used in other NEMO-based large-scale and global modelling experiments (Barnier et al., 2006). The water column





is discretized using 50 unevenly spaced vertical *z* levels with partial cells to fit the bottom depth shape. A 1/36° horizontal resolution was chosen for the NR in order to resolve the mesoscale in the ocean almost over the entire IBI domain (Hallberg, 2013).

An eddy resolving OGCM configuration was also used for the AR, but at a coarser spatial resolution to resolve the mesoscale
structures with a lower accuracy than the NR. In terms of spatial resolution, the difference between the two configurations is that the AR uses a 1/12° tripolar grid (ORCA12) and 75 vertical levels. A different spatial resolution between the NR and AR configurations was chosen to determine how assimilating high-resolution data into a coarser OGCM can contibute to increase the accuracy of the representation of the "true" mesoscale dynamics (given by the NR). In order to obtain independent results and quantify the impact of assimilating synthetic satellite altimetry observations from the NR in the AR, the two configurations
were initialized differently. Initial conditions in AR were obtained from a 7-years model spin-up (2002 - 2008) performed as a free run, forced by atmospheric forcings but without data assimilation.

The AR is forced by 3 h, 0.5° horizontal-resolution atmospheric re-analyses from the European Centre for Medium-Range Weather Forecasts (ECMWF-ERA-Interim, Dee et al., 2011). Atmospheric pressure and tidal potential (Lyard et al., 2006; Egbert and Erofeeva, 2002) are included in the model forcings. Lateral open boundary and initial conditions fields (Temperature,
Salinity, Velocities and Sea level) are obtained from the Mercator global ocean reanalysis (daily output) at 1/4° (GLORYS; Garric et al., 2018). As the atmospheric pressure forcing is not considered in the global reanalysis, the inverse barometer effects (e.g. Wunsch and Stammer, 1997) are computed from the ECMWF pressure fields and applied along the boundaries. Tidal harmonics were obtained from a 10-year free run simulation, performed using the same OGCM configuration (AR). On the other hand, NR is forced by the 3-h, 0.25° horizontal-resolution operational analyses from ECMWF and Mercator global
ocean analysis data are considered as initial and boundary conditions. In this configuration, tidal harmonics were obtained from the last version of the FES (Finite Element Solution) tide model (FES2014, Lyard et al., 2017). All the differences between the NR and AR configurations are listed in Table 1.

Starting from the same initial conditions, the OSSEs were performed from the 1st of January 2009 over almost 1-year time period, assimilating synthetic observations from different satellite constellations, in-situ temperature and salinity profiles and
SST maps.

## 2.2   Data assimilation scheme (DAS)

The impact study designed in this work was performed using an updated version of the data assimilation scheme developed at Mercator Ocean, called SAM2 (Système d'Assimilation Mercator V2), described by Lellouche et al. (2013a). In SAM2 the background error covariance matrix is based on a fixed collection of model anomalies. The anomalies are computed from a
numerical experiment and at each date they are given by the difference between the free run outputs and their temporal running mean. The aim is to obtain an ensemble of anomalies representative of the error covariances (Oke et al., 2008), which provide an estimate of the error on the ocean state at a given period of the year that is realistic with the climatological statistics.

In this study, we consider a 7-year free model run to obtain anomalies for Temperature (T), Salinity (S), zonal velocity (U), meridional velocity (V) and Sea-Surface Height (SSH). At the date of an analysis the anomalies are considered over a $\pm$ 60-





day temporal window and from the different years, resulting in a number of anomalies equal to $\sim 365$ for each given analysis. These anomalies are selected according to the season of the assimilation cycle to get a basis evolving consistently with the model climatology. Thus the background errors are not propagated by the dynamical model but evolve with the time as errors are based on anomalies which change at each analysis date. In this study, the anomalies were obtained considering 25-hour

averaged fields. The localization of the error covariance is performed assuming a zero-covariance beyond a distance defined as twice the local spatial correlation scale, which is about 80 km in the IBI region. The spatial correlation scales, estimated from IBI regional ocean re-analysis (Sotillo et al., 2015) , are also used to select the data around the analysis point.

The model correction (analysis increment) is a linear combination of these anomalies and depends on the innovation (observation minus model forecast equivalent as in Ide et al., 1997) and on the specified observation errors. This correction is

10 applied progressively over the assimilation cycle temporal window using an incremental analysis update (IAU; Bloom et al., 1996; Benkiran and Greiner, 2008) for an enhanced dynamical balance. In this study wide-swath altimetry data were obtained considering a $\sim$20-day repeat orbit with a 10-day subcycle (Gaultier et al., 2016a). Here we selected a 5-day assimilation cycle, because it seems appropriate regarding the half cycle of wide-swath data, and the ocean analysis was performed in the middle of the assimilation window.

A bias correction based on variational methods (3D-Var) is applied to the model's prognostic equations to correct large scale and slowly evolving errors in T and S diagnosed from the in-situ profile innovations. The model equivalents to SSH observations was computed considering a 25-hour average, to filter the tidal signal. Finally, with respect to the operational systems (Lellouche et al., 2013a), we have assimilated the full SSH signal instead of the sea level anomaly (SLA), as in Verrier et al. (2017). As proposed by Errico et al. (2013), the back-ground error and observation-error statistics are specified as in

the operational system for SST and in-situ observations, and the same quality control and data selection procedures are used either considering simulated or real data in order to obtain results in the OSSEs that can match with a real ocean analysis. A specific OSSE, not shown here, was performed for calibration purpose mimicking the present altimetric observing systems with a 3 $cm$ observation error prescribed on along-track observations, instead of 1 $cm$ chosen for OSSE1 (Section 3) presented in this paper. In the open ocean, the innovations statistics to the along-track observations in this OSSE and in the IBI system

assimilating real observations have a similar amplitude and patterns.

### 2.3 Simulation of observations

### 2.3.1 Sea-surface temperature (SST) and Temperature and Salinity profiles

To accurately evaluate the impact of satellite altimetry data on the ocean analyses and forecasts, the same synthetic observations of SST, T and S profiles were considered in all the experiments. We used daily averages of the NR to simulate the satellite SST

maps. One daily SST map is used during the 5-day assimilation cycle. Figure 2 shows an example of synthetic SST field to be assimilated for the 15/06/2009. Temperature and salinity profiles are extracted at the same points and the same dates as the real in-situ profiles found in the CORA3.2 data base from CORIOLIS data center (Cabanes et al., 2013). Figure 1, right panels, shows the number of T and S profiles available during 2009 over the IBI region.



### 2.3.2 Satellite altimetry data

In order to investigate the impact of different constellations of satellite altimeters, both conventional along track and wide-swath altimetry measurements were considered. Conventional altimetry data were derived from sampling the NR over the theoretical tracks of the satellite missions Jason 2, Cryosat 2 and Sentinel 3a, with a sampling frequency of 1 Hz. An observation white

noise of 1 cm rms was simulated and added to these pseudo-observations. Using the same approach, wide-swath data were derived considering a 20.9 days repeat orbit at a spatial resolution of of 7 km along and across the swath. In order to investigate the impact of multiple wide-swath altimeter missions, the data were derived simulating two wide-swath altimeters, hereafter Swath-1 and Swath-2, obtained considering a 10-day shift in the orbits of the simulated missions. Figure 2 shows the spatial coverage of simulated satellite altimetry data during 5 days (analysis window) considering conventional nadir (top left panel),

along Swath-1 (top central panel) and along Swath- 1 and 2 (top right panel) altimeter missions.

### 2.3.3 Wide-swath altimetry data

As previously mentioned, wide-swath altimetry observations were obtained sampling the NR in order to mimic wide-swath SSH measurements. A 7 km grid resolution was considered to be consistent with the horizontal resolution of AR configuration. Wide-swath altimetry measurements can be characterized by correlated and uncorrelated errors. A detailed description of

the wide-swath (SWOT-like) altimetry errors is given by several works in the literature (e.g. Esteban Fernandez et al., 2014; Dibarboure and Ubelmann, 2014; Gaultier et al., 2016a; Ubelmann et al., 2015; Ruggiero et al., 2016). Aware of the importance of a full characterization of the errors to exploit the information coming from wide-swath altimetry data (Ubelmann et al., 2017), as a first attempt to investigate their contribution to the ocean analyses and forecasts we focus only on the instrumental uncorrellated errors due to radar interferometer (thermal noise) and on their cross-track variability. Figure 2, bottom left panel,

shows the standard deviation of the random error obtained considering different radar interferometer configurations and an across swath horizontal resolution of 1 and 7 km. In this sense, it is important to notice that the error has marked spatial variability across the swath, reaching the highest values at the edges and the lowest near the inner part of the swath.

The measurement errors of the radar interferometer were defined in collaboration with TAS, contractor of radar altimeters for EO in Europe. In this study, the error due to a Ku-band Klystron Dual Receive Antenna (DRA) was considered and a

cross-track spatial resolution of 7 km was selected to be consistent with resolution of the simulated satellite altimetry data.

## 3 Experimental set-up: OSSEs

In this section we describe the set-up of the OSSEs performed to investigate the impact of wide-swath altimetry data once assimilated in an ocean monitoring and forecasting system in the IBI region.

First, a reference experiment, hereafter OSSE0, was realized considering the AR configuration but without any synthetic

observation. Other four different OSSEs have been carried out varying the type and number of altimeter missions considered. To evaluate how the constellation of nadir altimeters constrain the ocean analysis and forecast, an experiment was realized





considering exclusively conventional nadir altimeters (Jason-1, Cryosat-2, Sentinel-3a), hereafter OSSE1, considering an instrumental error in the order of 1 $cm$. A second experiment, hereafter OSSE2, was performed considering nadir altimeters and Swath-1 data to address the question about the impact of having in the future SSH measurements based on both nadir and wide-swath altimeter missions. As already mentioned, SWOT-like data have a temporal resolution which could not allow
to resolve correctly the evolution of mesoscale structures. In order to investigate the impact of the repetitivity of wide-swath altimetry data, in the experiment OSSE3 an additional wide-swath altimeter was considered, relative to OSSE2. This first series of OSSEs (OSSE2 and OSSE3) was performed assimilating wide-swath altimetry data simulated assuming a radar interferometer error that ranged between 0.8 $cm$ in the inner part of the swath and 2 $cm$ on the outer edges. The order of magnitude of the instrumental error was selected, in close collaboration with TAS, to consider less stringent noise requirements compared
to NASA/CNES SWOT mission. In particular, we considered an instrumental error four times larger than the error prescribed for the Ka-band Radar Interferometer (KaRIN) onboard the SWOT mission (TAS technical report). The bottom left panel in Figure 2, shows the across swath error obtained considering a spatial resolution both at 1 km (black line) and 7 km (orange line). In order to investigate the sensitivity of the ocean analysis and forecasting system to the error of wide-swath altimetry instrument, a dedicated OSSE, hereafter OSSE4, was realized considering a satellite constellation as in OSSE3 but assuming
a radar interferometer error which has an halved order of magnitude (0.4 - 1 $cm$) with respect to the other OSSEs (2 x KaRIN error). The experimental set-up used in this study is listed in Table 2.

## 4   Impact on sea-level analyses and forecasts

In this section we compare the SSH in the different OSSEs, with the "truth" SSH given by the NR. Figure 3 shows the SSH variance in the NatRun, computed over the period February – December 2009. The IBI region is characterized by relatively
steep slope separating the deep ocean from the shelf (Maraldi et al., 2013). On the continental shelves, the barotropic component of the SSH has a dominant signature. Preliminary findings of this work showed that this kind of variability was captured also in the reference simulation (OSSE0), without data assimilation, while in the deeper areas of the ocean it was not accurately reproduced and the variance of the error was larger than 50 % of the variance of the NR (not shown). To evaluate the results we considered $VAR^*$ defined (in terms of percentage) as:

$$25 \quad VAR^* = 100 \, \frac{VarError(OSSE_k)}{Var(NR)} \tag{1}$$

where $VarError$ is the variance of the error obtained comparing a given OSSE with the NR, $k$ refers to the $kth$ experiments and $Var(NR)$ is the variance of the signals in the NR.

In the following part of this Section, the results are presented also in terms of contribution of wide-swath altimetry data to the reduction of the error (ER*), both in the ocean analysis and forecast. In order to asses the impact of future satellite missions





with respect to the current constellation of nadir altimeters, the ER* is defined as the percentage decrease of the error with respect to the OSSE1, i.e.

$$ER^* = 100 \; \frac{VarError(OSSE1) - VarError(OSSE_k)}{VarError(OSSE1)} \qquad (2)$$

where $VarError$ is defined as in Eq.(1), and $k$ refers to the $kth$ experiments, with $k = 2, .., 4$. A value of 50 % means that the variance of the error in the $kth$ experiment has halved with respect to OSSE1.

Wide-swath altimetry data are expected to provide a significant contribution to resolve the mesoscale and sub-mesoscale variability in the ocean, which can have different spatial scales defined by the Rossby deformation radius in the different regions of the ocean. As shown in Hallberg (2013), over the shelf in the Celtic and North Seas an higher horizontal resolution (1/50°) is needed to resolve the first baroclinic instability mode than the one used in the OSSEs (1/12°), which instead results as being suitable for resolving the ocean dynamics in the Atlantic. Thus, in our configurations the contribution of altimetry data in the OSSEs is expected to be more evident in the open ocean than over the continental shelf. In order to take into account the effects of the SSH barotropic component and of the spatial resolution over the shelves, the results in the OSSEs were evaluated both over the entire IBI domain and considering the ocean areas with a bathymetry deeper than 200 metres (Table 3), which is the isobath that typically represent the separation between continental slope and shelf in the bathymetry adopted by the OGCMs used in the OSSEs (Maraldi et al., 2013).

In the experiment which considers only nadir altimeters (OSSE1) the error of the SSH in the ocean analysis represents $\sim$ 20-30 % of the variance of the SSH signal in the NR ($VAR^*$ in Table 3). One of the most significant results of this work is about the impact of a constellation of satellites which combines nadir altimeters and one wide-swath instrument (OSSE2). Comparing the SSH in the ocean analysis with the "truth" data, the results showed a significant positive impact in the system and a reduction of the variance of the error ($ER^*$) up to order of $\sim 30\%$ was observed, with respect to the error observed assimilating the data of a constellation of three conventional altimeters (Figure 4). It is interesting to notice that taking out the shelf areas the variance of the error increased in OSSE1, while in the experiments with wide-swath data had a similar order, with respect to the error observed over the whole domain ($VarError$ in Table 3), underlining the impact of future measurements in the open ocean. As a consequence, the $ER^*$ in OSSE2 increased up to the order of $\sim 35\%$ considering the ocean areas with a bathymetry deeper than 200 metres, with respect to OSSE1 over the same spatial domain.

In terms of $VAR^*$, OSSE2 also shows a 6 % reduction with respect to OSSE1. A larger impact was observed in the ocean forecast (Figure 5), where the $ER^*$ increased up to the order of $\sim$ 20 %, considering the last (5th) day of ocean forecast (Table 4). When nadir altimeters were combined with a constellation of two wide-swath altimeters (OSSE3), the impact on the ocean analysis and forecasting system was even more significant, with an $ER^*$ in the order of 40-45 % considering the entire IBI domain. The impact of the repetitivity of wide-swath data can be noticed comparing the results in OSSE3 and OSSE2. A difference of $\sim$ 10 %, in terms of $ER^*$, was observed considering the data of two wide-swath missions in OSSE3, with respect to those obtained assimilating a single wide-swath altimeter in OSSE2. The positive impact observed was mainly due to improvements in the representation of SSH variability in the Bay Biscay and in the occurrence of the Azores current (Figure



6). Inter-comparing OSSE2 and OSSE3, it is also interesting to notice the impact on the forecasting of the SSH. A constellation of two wide-swath altimeters shows a significant $ER^*$ in the ocean forecast till the 5th ( $\sim 28$ %) day of forecast, which almost corresponds to the error reduction observed at the 1st day of forecast considering only a single wide-swath altimeter (29 %).

Looking at the sensitivity of the system to the instrumental error of wide-swath data, a smaller radar interferometer noise was used in OSSE4 ($0.4 - 1 \, cm$), relative to the error used in other OSSEs. The results showed a larger $ER^*$, in the order of 7-8 % (Figure 6), with respect to the experiment which consider the same constellation of satellite altimeters (OSSE3) but a larger instrumental error ($0.8 - 2 \, cm$). In particular in OSSE4 was observed the largest $ER^*$ ( $\sim 50$ %) and the errors represent the smallest portion (10-15 %) of the "observed" ocean variability ($VAR^*$).

The results of the impact of wide-swath altimetry data on the representation of the SSH in the ocean analysis are summarized
in Table 3.

## 5 Spectral analysis and coherence

In this Section we describe the results obtained in the OSSEs in terms of the representation of the ocean dynamics, looking at the errors over different spatial scales of variability.

The impact of wide-swath altimetry data in the OSSEs was evaluated by a power spectra comparison, considering an open
ocean area representative of the North Atlantic Drift (19 W°, 10 W°; 46°N, 55°N). This region is defined as an intermediate mesoscale variability region in the global ocean (Garçon et al., 2001) and represents one of the regions which show the highest variability of the ocean circulation within the IBI domain. In particular here we focus on the impact of wide-swath altimetry on the SSH at the different spatial scales. A wavelength window between 400 and 12 km was selected in order to clearly represent the energy content of the SSH signal in the sub-domain, given the spatial resolution of the OGCM used to perform the OSSEs.
The analysis of spectra in a variance preserving form (Thomson and Emery, 2014) is shown in Figure 7. The power spectra of the error (left panel) clearly show the differences among the OSSEs. In order to evaluate also the impact of nadir altimeters, here the reduction of the error at the different wavelength ($ER_{spec}$) is defined as the percentage decrease of the error with respect to OSSE0 (Table 5).

Considering nadir altimeters (blue curve) the impact on the ocean analysis can be noticed at spatial scales down to 100 km
and an $ER_{spec}$ in the order of 60 % was observed at wavelengths between 100 and 200 km. Significantly larger contributions were observed considering wide-swath altimetry data ($ER_{spec}$ up to > 80 %). Inter-comparing the results in OSSE1 with those obtained in the other experiments (Figure 7, left panel), the reduction of the error at these length scales ranged between 40 and 55 %. This is agreement with Dufau et al. (2016) who, investigating the resolution capability of present and future altimetry missions, observed that wide-swath altimetry will provide an unprecedented insight into the mesoscale ocean dynamics, with
respect to along-track data. Here it is also interesting to notice the difference among OSSE2 and OSSE3 (purple and red lines), which show the impact of the repetitivity of the SSH measurements once a second wide-swath altimeter was considered within the constellation of altimeters.



The contribution of wide-swath altimetry data was also significant at spatial scales smaller than 100 km, which was not the case considering only nadir altimeters. In particular considering wavelengths between 50 and 100km wide-swath altimetry data showed the largest contribution, with respect to along-track data. At these spatial scales the $\mathrm{ER}_{spec}$ in OSSE1 was the lowest (10 %) observed in the spectral analysis. Combining nadir altimeters with one wide-swath instrument (OSSE2), the results

showed an $\mathrm{ER}_{spec}$ in the order of 28 %. The introduction of a second wide-swath altimeter in OSSE3, assuming the same instrumental error used in OSSE2, showed a small further reduction of the error (30 %). On the other hand, considering a lower radar interferometer in OSSE4 enhanced the system to better resolve the ocean dynamics at these scales of variability and the largest $\mathrm{ER}_{spec}$ ($> 38$ %) was observed, underlining the sensitivity of the system to the error of wide-swath measurements.

A coherency analysis (Thomson and Emery, 2014) was also performed to investigate the reliability of the SSH signal in the

OSSEs at the different spatial scales, with respect to the NR (Figure 7 right panel). Spectral coherence is typically defined as the correlation between two signals as a function of wavelength (Ubelmann et al., 2015; Ponte and Klein, 2013; Klein et al., 2004). The spectral coherence between the SSH signals in the NR and in the OSSEs is defined as follow:

$$C_{spec} = \frac{Cr_s(NR, OSSE_k)}{S(NR)\,S(OSSE_k)} \tag{3}$$

where $Cr_s$ and $S$ represent the cross-spectral density and spectral density, respectively, of the signals and $k$ refers to $kth$

experiment.

Differences among the OSSEs in terms of spectral coherence can be noticed down to spatial scales between 50 and 100 km. The assimilation of wide-swath altimetry data increased significantly the coherence between the SSH signals, with respect to the experiment which considers only nadir altimeters. At the large scale (200-400 km) the coherence between the SSH signals was fairly high ($> 0.8$) in all the OSSEs. Considering the coherence values at relevant spatial scales it is possible to inter-compare

the results in each experiment, as shown in Table 5. In particular, a different coherency value observed at the same length scales in the OSSEs provide evidences about the increased (decreased) level of reliability obtained considering (neglecting) a given observing system. At spatial scales between 100 and 200 km, the coherence increased by 20 % considering one wide-swath and the nadir altimeters (OSSE2) relative to OSSE1. Similar values were observed considering two wide-swath altimeters in OSSE3. Looking at the sensitivity of the system to the radar interferometer error, the observing system designed in OSSE4

had the most significant impact on the spectral coherence in the ocean analysis at these spatial scales, relative to OSSE1 ($> 20$ %). At the small scales ($< 90$ km) the coherence was lower than 0.4 in all the experiments. Aware of the limited significance of low coherency values ($< 0.5$), here we compare the results obtained in the OSSEs to obtain a qualitative evaluation of the impact of the forthcoming altimeters on the spatial scales smaller than 100 km. At these wavelengths the coherence in OSSE1 was always lower than in the other OSSEs and the system was sensitive to the repetitivity of the wide-swath measurements

considered: introducing the data of one (OSSE2) and two (OSSE3) wide-swath altimeters the coherence of the SSH signals in the ocean analysis increased accordingly. The difference between the results of OSSE3 and OSSE4 was not significant ($C_{spec}$ ranged between 0.2-0.4), even though an higher coherency of the SSH signals due to "accurate" wide-swath altimeters (Figure 7 green line) was qualitatively noticed down to spatial scales smaller than 70 km.





These results can be qualitatively extended to the ocean forecast considered over the same spatial domain (not shown). The results of the spectral analysis, in terms of error and coherency, are summarized in Table 5.

## 6   Impact on velocity fields

In this Section we compare the contribution of wide-swath altimetry to the representation of the ocean circulation in the IBI region considering ocean analyses. In order to investigate the reliability of the ocean circulation obtained in the different experiments, a comparison of the zonal and meridional currents was carried out, both at the surface and in the water column (Table 6). Figure 8 shows the variance of the error obtained comparing the surface zonal velocities in the OSSEs with the same field in the NR during a 2-month period (August-September 2009). In the Figure, the first panel shows the variance of the error obtained considering conventional nadir altimeters (OSSE1) and large errors were observed in the Atlantic. When wide-swath altimetry data were considered, a significant reduction of the error was observed, particularly evident in the North-Atlantic Drift, in the occurrence of the Azores Current and in the Bay of Biscay. In particular, considering a constellation of three nadir and one wide-swath altimeters (OSSE2) an $ER^*$ up to the order of 20 % was observed, with respect to the error observed considering only conventional altimeters. A larger impact on the ocean circulation at the surface ($\sim$ 28 %) was observed considering a constellation of three nadir and two wide-swath altimeters (OSSE3). Considering more accurate wide swath altimeters (OSSE4), the results showed a further reduction of the error in the domain ($\sim$ 35 %), mainly due to an improved representation of the North Atlantic Current. Small positive $ER^*$ (2-3 %), were observed considering the ocean surface currents only in the deep ocean (> 200 m). A coherency analysis performed considering the ocean currents (zonal and meridional) at the surface (not shown), confirmed the results observed considering the SSH fields. The results obtained for the zonal currents at the surface can be extended also to the meridional currents, and in general to the ocean currents in the water column, as shown in Figure 9. The results of the comparison of the ocean currents in the OSSEs are summarized in Table 6.

## 7   Summary and conclusions

The contribution of wide-swath altimetry data on ocean analyses and forecasts was evaluated in the IBI region during a 1-year period (2009) by means of OSSEs. Five different experiments (4 OSSEs and 1 reference simulation OSSE0) were designed simulating different constellations of satellites, composed by nadir and wide-swath altimeters, and the results compared with the NR. OSSE1 is representative of the constellation of altimeters at present and considers simulated data of three nadir altimeters (Jason2, Cryosa2 and Sentinel3a). Dedicated experiments performed for calibration purpose, prescribing a 3 $cm$ observation error on along-track observations to mimic the present altimetric observing systems, showed consistent results with those derived considering real data in OSEs addressed to evaluate the impact of multiple along-track altimeters in CMEMS systems.

OSSE2 differs from OSSE1 due to the introduction of a wide-swath altimeter in the satellite constellation. The impact of the repetitivity of swath-swath measurements was investigated in OSSE3, considering a further wide-swath mission, with respect



to OSSE2. The sensitivity of the system to the wide-swath radar interferometer error was investigated in OSSE4, considering a lower instrumental error (0.4 - 1 $cm$) with respect to that used in the other experiments.

An initial result of this work regards the reliability of the SSH signals that can be obtained considering different constellations of altimeter missions. Wide-swath SSH measurements had a major impact to reduce the error in the ocean analyses and

forecasts. A constellation of two ("accurate") wide-swath altimeters allow reducing the variance of the SSH errors by more than 50 % with respect to three conventional nadir altimeters, mainly due to an improved representation of the ocean mesoscale variability in areas of main ocean currents occurring in the IBI region (e.g. North Atlantic and Azores Currents). Looking at the SSH ocean forecasts, the most significant results were obtained investigating the impact of two wide-swath altimeters. In particular the $ER^*$ given by a constellation of three nadir altimeters and one wide-swath (3N+1S) mission at the first day of

forecast ($\sim$ 30 %) is comparable with the $ER^*$ obtained at the last (5th) day of forecast considering a second wide-swath altimeter (3N+2S) in the satellites constellation. The last aspect could have a strong implication for extending the temporal window in the ocean forecast. The ocean analysis and forecasting system used to perform the OSSEs was sensitive to the repetitivivity of wide-swath measurements and to the wide-swath altimeters instrumental error. The most significant results were obtained considering a constellation of three nadir and two "accurate" wide-swath altimeters, which contributed to reduce

the error in the ocean analysis up to the $\sim$ 10 % of the "observed" ocean variability ($VAR^*$) in the (ocean) analysis.

Evaluating the SSH signals in the OSSEs by power spectra comparison and coherence analysis with the NR in one of the most energetic sub-regions in the IBI domain (North Atlantic Drift), the results showed that wide-swath altimetry data significantly contribute (40-50 %) to resolve the ocean dynamics due to the mesoscale variability. This is agreement with Dufau et al. (2016) who observed that wide-swath altimetry will provide an unprecedented insight into the mesoscale ocean dynamics, with respect

to along-track data. A reduction of the error and an higher coherency of the SSH signals can be noticed down to wavelengths < 100 km considering two accurate wide-swath altimeters within a future satellite constellation, relative to the present. Similar results were observed also considering the ocean forecast and performing the same analysis on the ocean currents at the surface. The information provided by wide-swath data at the surface is propagated, through the error covariances, also in the vertical and has a considerable impact on the ocean circulation both at the surface and in the water column. Looking at the zonal currents

and meridional currents at the surface and at depth, the error in the analysis was significantly reduced (30 %) considering two wide-swath instruments in the satellite constellation, relative to nadir altimeters. In particular, a reduction of the error down to $\sim$ 10 % was observed on the ocean currents (up to 100 m) considering a constellation of two "accurate" wide-swath altimeters, with respect to wide-swath instruments with a larger observational error.

As already mentioned, the system used to asses the impact of the wide-swath altimetry data is sensitive to the instrumental

(uncorrelated) errors due the radar intereferometer. Satellite constellations of nadir and "accurate" wide-swath altimeters would have in the future a dramatic impact to constrain CMEMS ocean analysis and forecasting systems and their applications.

This study represents a first effort to quantify the impact of a constellation of wide-swath altimeters on ocean analyses and forecasts. In the future, OSSEs should be performed in regions characterized by high mesoscale variability (i.e. western boundary currents) to better assess the impact of measurements errors in regions with large signal to noise ratio. The sensitivity

of results to the model spatial resolution should be assessed by performing OSSEs with a fully eddy resolving ocean model



(e.g. 1/36°). Finally, a full and accurate characterization of wide-swath altimeters error spectrum (Ubelmann et al., 2017) would allow the design of highly realistic OSSEs, improving the definition of the error covariances required to combine wide-swath altimetry data and OGCMs through data assimilation techniques.

*Acknowledgements.* This work was supported by the European Space Agency (ESA) study on the potential role of wide-swath altimetry for

5  the evolution of the European Union Copernicus Programme (contract ESA-Mercator Ocean n° 4000117621/16/NL/FF/gp).



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





**Figure 1.** Temperature and Salinity simulated data. Left panel: sea-surface temperature from the NR (15th June 2009). Right panels: number of temperature (top) and salinity (bottom) data during the year 2009 (corresponding to the database CORA3.2 in-situ data positions).





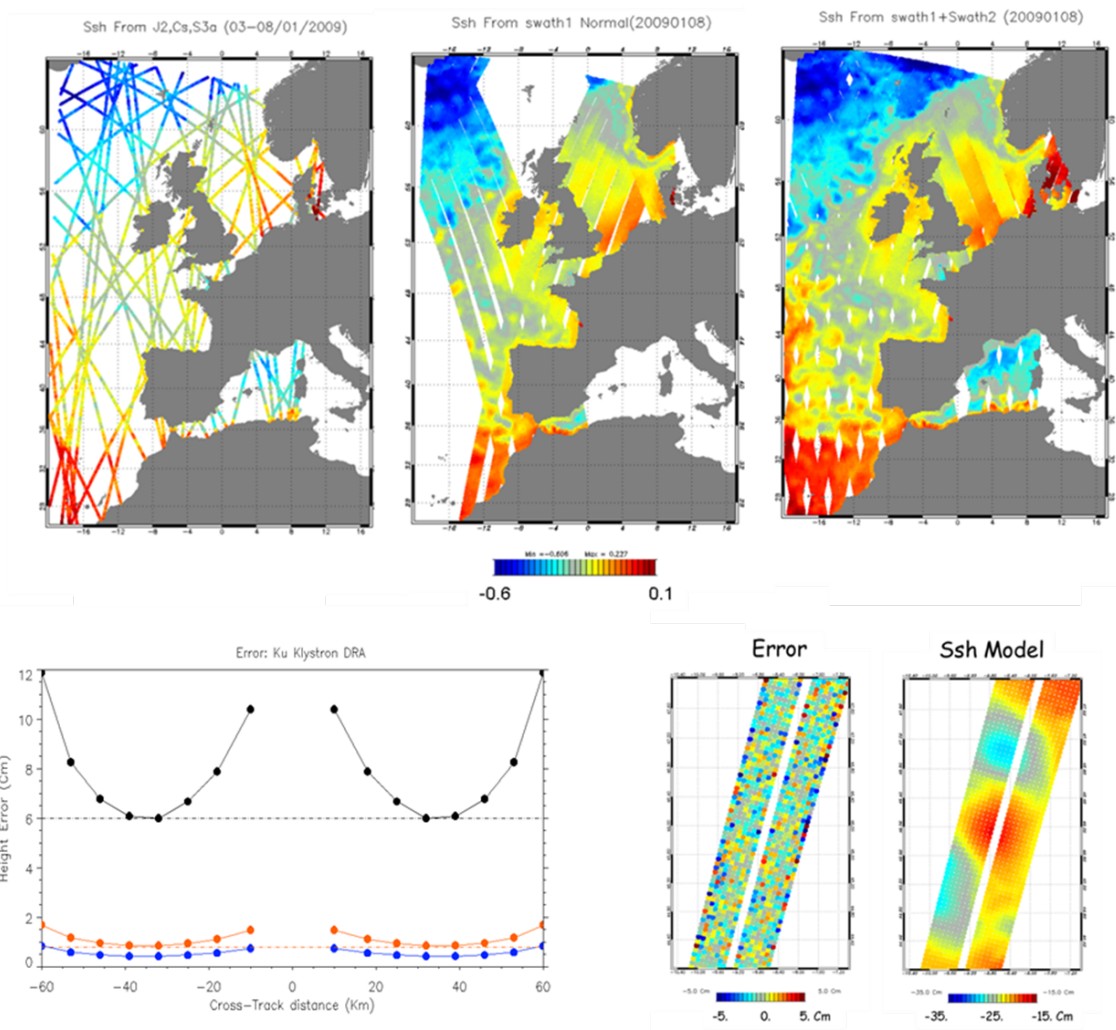

**Figure 2.** Satellite altimetry spatial coverage and wide-swath interferometer error. Top panels: satellite altimetry spatial coverage during one assimilation cycle (5 days); from the left: Jason2, Cryosat 2 and Sentinel 3a; Swath-1; Swath-1  Swath-2. Bottom panels: across swath error. Left: the curves displayed show the wide-swath instrumental error used to perform the experiments OSSE2 and OSSE3, considering an across swath horizontal resolution of 1 km (black line) and 7 km (orange line). The blue curve shows the instrumental error used in OSSE4. Right: along swath error given by the radar interferometer noise and along swath simulated data.





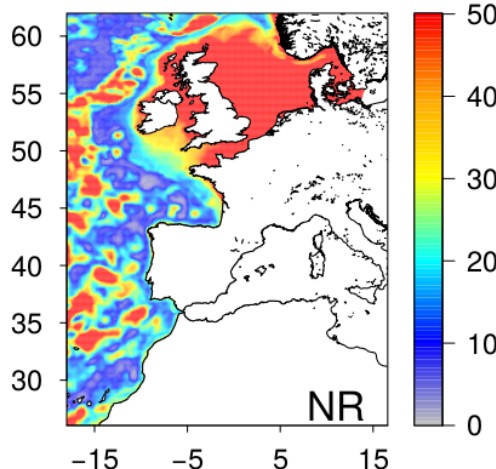

**Figure 3.** SSH variance in the NR over the period February-December 2009. Values expressed as $[cm^2]$.

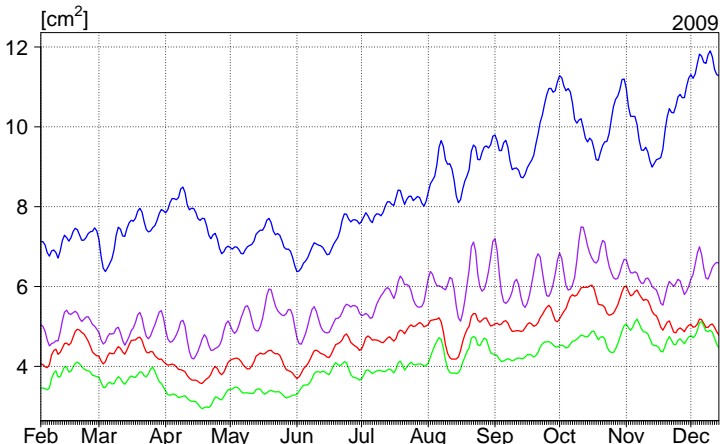

**Figure 4.** Temporal evolution of the SSH error variance in the ocean analysis over the period February-December 2009. Results obtained comparing the SSH of the ocean analysis in the experiments OSSE1 (blue lines) and OSSE2 (purple line), OSSE3 (red line) and OSSE4 (green line) with the data of the NR considering the ocean areas with a bathymetry deeper than 200 metres; y-axes expressed as $[cm^2]$. OSSE0 (not shown) ranged between 25 and 35 $cm^2$ .



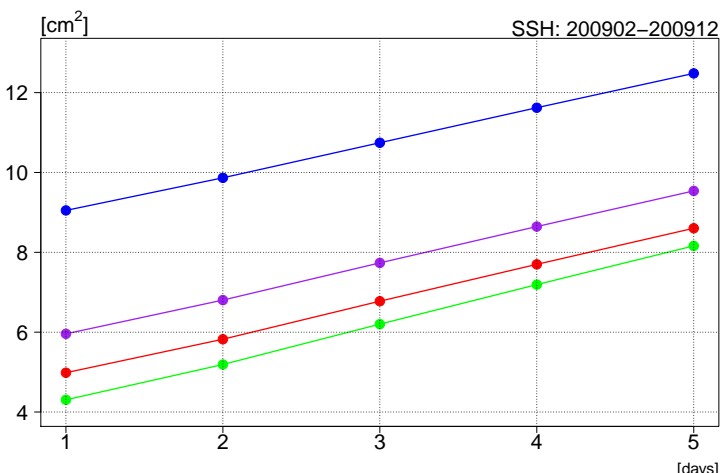

**Figure 5.** Variance of the error for each day of forecast (5 days) considering the SSH, in regions with a bathymetry deeper than 200 metres, over the period from Frebruary to December 2009. The legend of the OSSEs is as in Figure 4; y-axes expressed as $[cm^2]$.




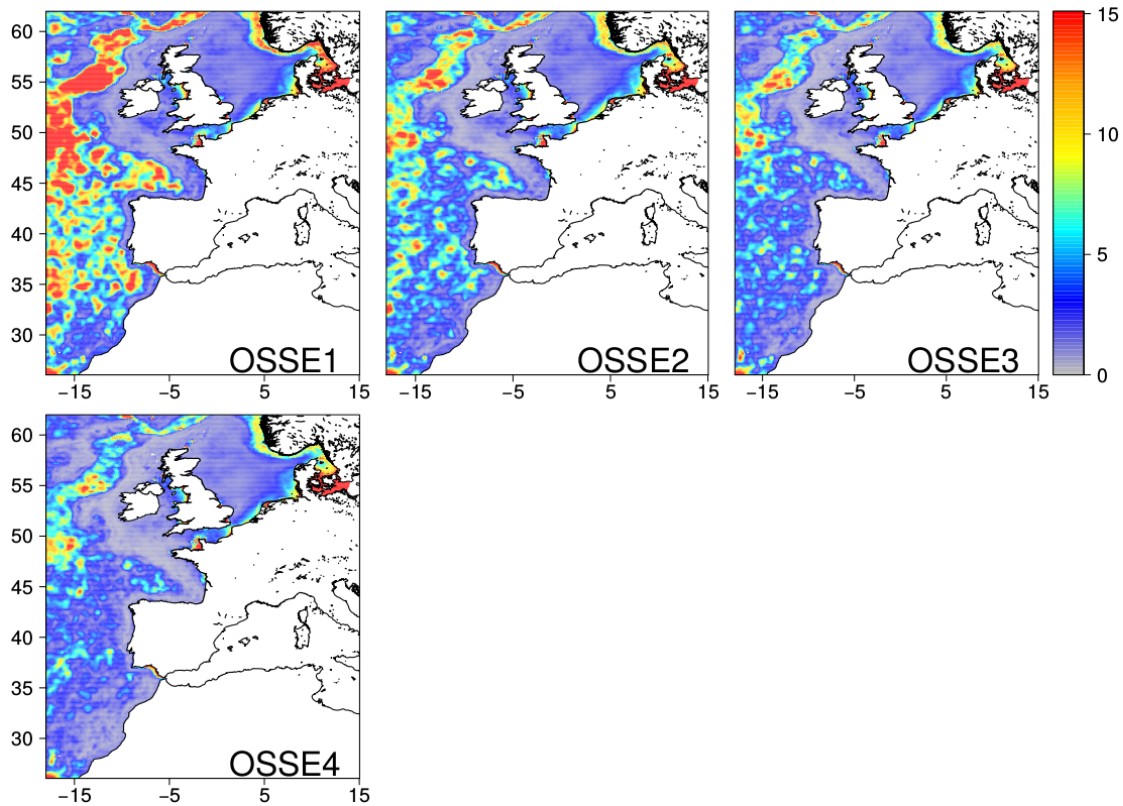

**Figure 6.** Maps of the variance of SSH error in the ocean analysis over the IBI domain during the period February-December 2009. The results are obtained comparing the SSH of the NR with the ocean analysis of the experiment OSSE1 (top left), OSSE2 (top central), OSSE3 (top right), OSSE4 (bottom left). Values expressed as $[cm^2]$.



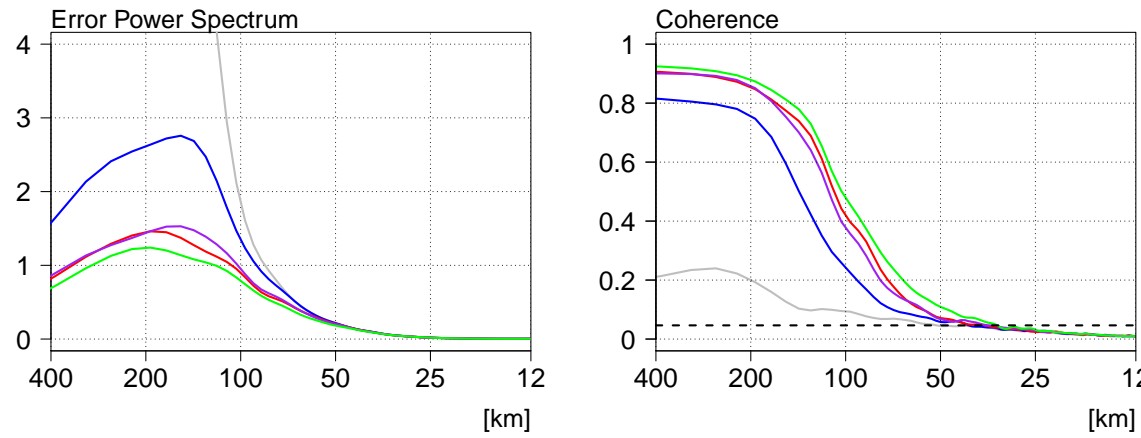

**Figure 7.** Spectral Analysis of the SSH signals in the OSSEs considering an open ocean region (-19°W, -10°W; 46.N, 55°N) representative of the North Atlantic Drift, during the period February-December 2009. In the panels the results for the experiments OSSE0 (gray lines), OSSE1 (blue lines), OSSE2 (purple lines), OSSE3 (red lines) and OSSE4 (green lines), are shown at the spectral window between 400 and 12 km. Left panel: power spectra of the SSH error, with respect to the NR; the spectra are shown in a variance preserving form [$cm^2$]. Right panel: spectral coherence in the OSSEs with respect to the NR; black dashed line shows the 95 % confidence interval (Thomson and Emery, 2014).




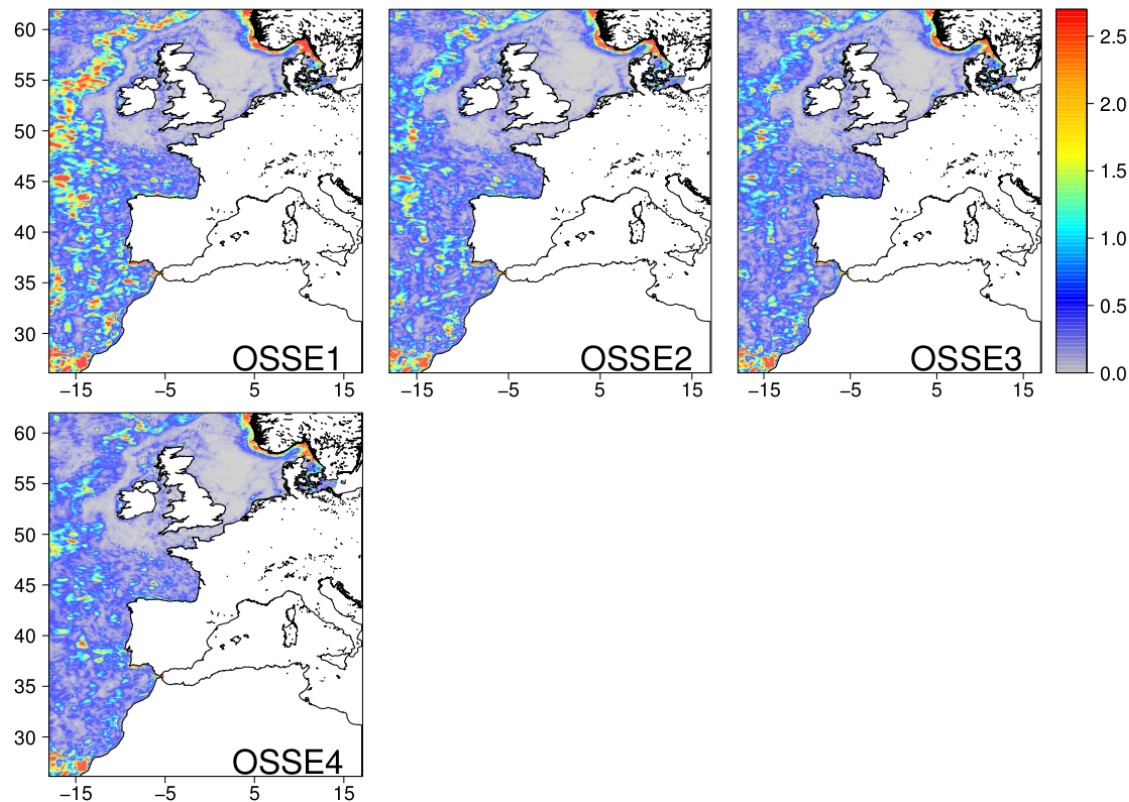

**Figure 8.** Ocean zonal currents at the surface: variance of misfits in the ocean analysis over the IBI domain during the period August-September 2009. The results are obtained comparing the zonal currents of the "truth data" with the ocean analysis in the experiment OSSE1 (top left), OSSE2 (top central), OSSE3 (top right), OSSE4 (bottom right) . Values expressed as [ $10^{-2}\ m^2\ sec^{-2}$ ].




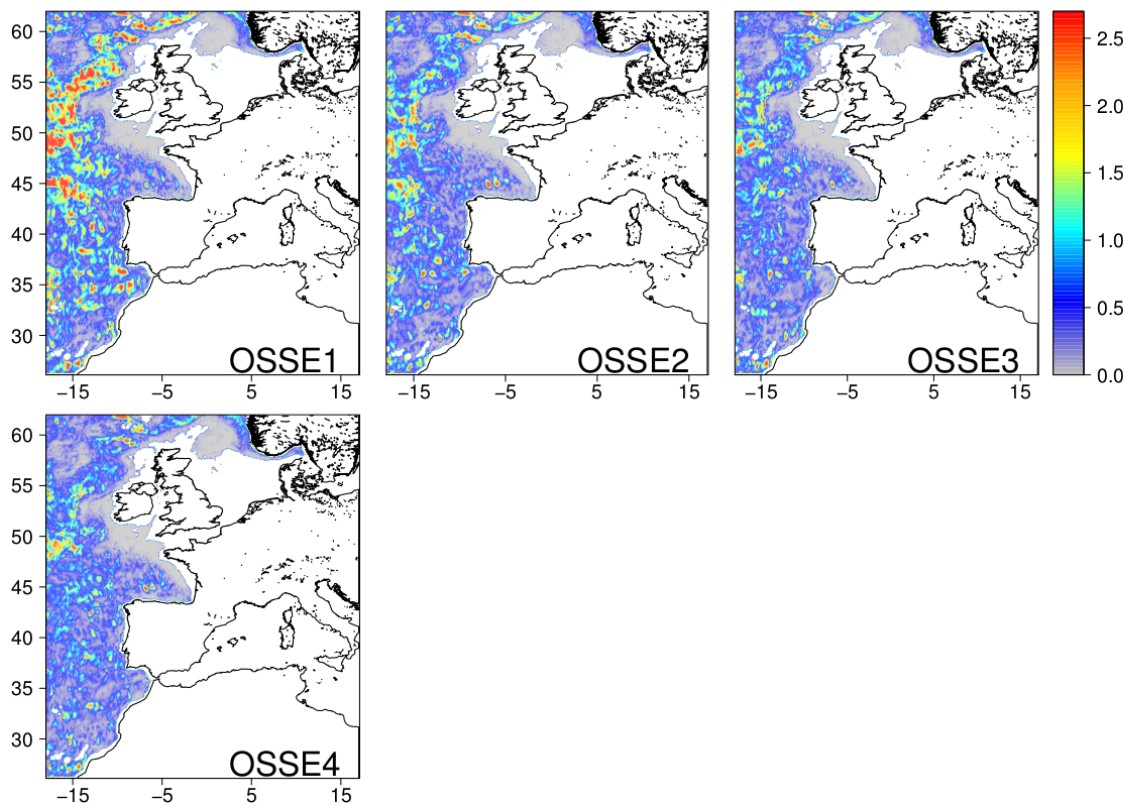

**Figure 9.** As in Figure 8 but considering ocean meridional currents at 100 metres depth.



**Table 1.** The nature run (NR) and assimilated run (AR) configurations. The rows show the OGCM configurations used to obtain the nature run (NR) and the assimilated run (AR). The columns show the OGCM used, the horizontal resolution (GRID), initial (IC) and boundary (BDC) conditions and the atmospheric forcings (ATM) considered in each configuration.

|    | OGCM | GRID | IC & BDC | ATM |
|----|------|------|----------|-----|
| NR | NEMO v3.6 | 1/36 ° | Global Analysis (Lellouche et al., 2013b); Tides (FES; Lyard et al., 2017) | ECMWF Analysis |
| AR | NEMO v3.6 | 1/12 ° | Global Re-analysis (GLORYS; Garric et al., 2018); Tides (IBI) | ECMWF Era-Interim (Dee et al., 2011) |

**Table 2.** OSSEs experimental set-up. The rows show the name of the relevant experiment, while columns detail the observations considered in the analysis. Column 1 shows the nadir altimeters considered in the OSSEs: Jason2 (J2), Cryosat 2 (C2), Sentinel 3a (S3). Last Column shows the the instrumental error (expressed as $cm$) used to simulate the altimetry data; OSSE2-OSSE4: range of across swath errors considered for the wide-swath altimetry data. In the round brackets: reference simulation without data assimilation (Free); three nadir altimeters (3N); one wide-swath (1S); two wide-swath (2S); Lower radar interferometer Error (LE). In the squared brackets: order of magnitude of wide-swath altimetry error selected in the OSSEs, with respect to the error of KaRIN instrument (NASA/CNES SWOT mission) .

|  | J2,C2,S3 | Swath-1 | Swath-2 | T&S | SST | Error |
|--|----------|---------|---------|-----|-----|-------|
| **OSSE0 (Free)** | | | | | | |
| **OSSE1 (3N)** | YES | | | YES | YES | 1 |
| **OSSE2 (3N+1S)** | YES | YES | | YES | YES | 0.8-2 [4 x KaRIN] |
| **OSSE3 (3N+2S)** | YES | YES | YES | TES | YES | 0.8-2 [4 x KaRIN] |
| **OSSE4 (3N+2S+LE)** | YES | YES | YES | YES | YES | 0.4-1 [2 x KaRIN] |

**Table 3.** Ocean Analysis statistics considering the SSH fields during the year 2009. Column 1: ocean analysis variance of error with respect to the NR (VarError $[cm^2]$). Column 2: reduction of the variance of the error in the ocean analysis (ER [%]), with respect to OSSE1 (3N). Column 3: ratio among the variance of the error in ech experiment and the variance of the SSH signal in the NR (VAR* [%]). Column 4-6: as in Columns 1-3, but excluding the shelves areas shallower than 200 metres (e.g.: ER $_{>200m}$ [%] ). In the round brackets: as in Table 2.

|  | VarError $[cm^2]$ | ER [%] | VAR* [%] | VarError $_{>200m}$ $[cm^2]$ | ER $_{>200m}$ [%] | VAR* $_{>200m}$ [%] |
|--|-------------------|--------|----------|------------------------------|-------------------|---------------------|
| **OSSE1 (3N)** | 7.7 | - | 19 | 8.5 | - | 34 |
| **OSSE2 (3N+1S)** | 5.4 | 30 | 13 | 5.6 | 34 | 22 |
| **OSSE3 (3N+2S)** | 4.7 | 39 | 12 | 4.7 | 45 | 19 |
| **OSSE4 (3N+2S+LE)** | 4.2 | 46 | 10 | 4 | 53 | 16 |



**Table 4.** SSH ocean forecast error statistics. Column 1-3: reduction of the variance of error in the ocean forecast ($\mathrm{ER}_f$), with respect to OSSE1 (3N) at the 1st, 3rd, and 5th day of forecast. Columns 4-6: as in Columns 1-3, but excluding the shelf areas shallower than 200 m (e.g. $1st_{>200m}$). The legend of the OSSEs is as in Figure 4. Statistics obtained considering the period February-December 2009. Values expressed as a percentage [%].

|  | $\mathbf{ER}_f\mathbf{1st}$ | $\mathbf{ER}_f\mathbf{3rd}$ | $\mathbf{ER}_f\mathbf{5th}$ | $\mathbf{ER}_f\mathbf{1st}_{>200m}$ | $\mathbf{ER}_f\mathbf{3rd}_{>200m}$ | $\mathbf{ER}_f\mathbf{5th}_{>200m}$ |
|---|---|---|---|---|---|---|
| **OSSE2 (3N+1S)** | 29 | 25 | 21 | 34 | 28 | 24 |
| **OSSE3 (3N+2S)** | 39 | 33 | 28 | 45 | 37 | 31 |
| **OSSE4 (3N+2S+LE)** | 45 | 37 | 31 | 52 | 42 | 35 |

**Table 5.** Spectral analysis. Columns 2-4: error reduction (%) with respect to OSSE0 (control simulation) at different spatial scales. Columns 5-7: spectral coherence (0.8-0.4) in the OSSEs; the values show the spatial scale (expressed as kilometres) at which the coherence falls below 0.8, 0.6, 0.4.

|  | $\mathbf{ER}_{spec}$ [%] | | | $\mathbf{C}_{spec}$ | | |
|---|---|---|---|---|---|---|
|  | 200-400 $km$ | 100-200 $km$ | 50-100 $km$ | 0.8 | 0.6 | 0.4 |
| **OSSE1 (3N)** | 79 | 64 | 10 | 280 | 155 | 125 |
| **OSSE2 (3N+1S)** | 89 | 79 | 28 | 170 | 125 | 105 |
| **OSSE3 (3N+2S)** | 89 | 81 | 30 | 165 | 120 | 95 |
| **OSSE4 (3N+2S+LE)** | 91 | 84 | 38 | 150 | 115 | 90 |

**Table 6.** Zonal (U) and meridional (V) currents error reduction (ER), with respect to nadir altimeters only (OSSE1). Columns 1, 3-4: reduction in the ocean analysis considering the zonal currents at surface ($Us$), at 100m ($U100m$) and at 300m ($U300m$). Column 2: as in Column 1 but excluding the shelves areas shallower than 200 m ($Us_{>200m}$). Columns 5, 7-8: as in Colums 1-3 but considering meridional currents. Columns 6: as in Column 2 , but considering meridional currents ($Vs_{>200m}$). Values expressed as a percentage [%]. In the round brackets: as in Table 2.

|  | $\mathbf{ER}_{Us}$ | $\mathbf{ER}_{Us_{>200m}}$ | $\mathbf{ER}_{U100m}$ | $\mathbf{ER}_{U300m}$ | $\mathbf{ER}_{Vs}$ | $\mathbf{ER}_{Vs_{>200m}}$ | $\mathbf{ER}_{V100m}$ | $\mathbf{ER}_{V300m}$ |
|---|---|---|---|---|---|---|---|---|
| **OSSE2 (3N+1S)** | 21 | 23 | 23 | 23 | 23 | 25 | 24 | 24 |
| **OSSE3 (3N+2S)** | 28 | 30 | 29 | 26 | 29 | 31 | 31 | 29 |
| **OSSE4 (3N+2S+LE)** | 35 | 37 | 34 | 30 | 38 | 41 | 36 | 32 |