# Peer review of "Contribution of future wide swath altimetry missions to ocean analysis and forecasting"

_Ocean Science, 2018_

## Referee Comment (RC1) · Anonymous Referee #1 · 9 Jul 2018

**1   General comments**

There are currently 3 nadir altimeters in orbit, which provide along track measurements of Sea-Surface Height (SSH). However, due to their narrow band nature, and the repeat times, the gaps between tracks are large compared to the ocean mesoscale – thus only limited spatial and temporal coverage is available. Wide swath altimeters are being developed to which would provide much greater SSH data.

This paper looks at the implication of such data being available to constrain model solutions for the IBI region. They note that similar studies are needed for the global scale, and for other regions.

[Figure]

They run a series of Observations Synthetic Simulation Experiments to assess the impact of assimilating the current constellation of altimeters, and future wide band altimeters. I found the experimental design satisfying, and analysis convincing.

**2  Specific comments**

I found the paper well thought out, clear and useful, and would recommend it for publication with corrections.

I think it had scientific significance, scientific quality. The presentation quality is also quite good, although I think it needs to be proof read by another native-English speaker, as there are sections that are incorrect or unclear. I think they need to reproduce the graphics with a better colour-map – jet/rainbow is very misleading.

I thought there should be a bit more description of the altimeters and of the ocean forecasting/analysis/reanalysis system.

I think there could be a little more discussion of what the statistics mean physically, in terms of location, mechanism etc.

I have outlined these specific comments here.

Page 4: End of introduction. Later in the paper when you start talking about forecast errors, I realised that it was not a reanalysis study – I wonder if this should be further clarified in the introduction. . . perhaps a sentence of two describing the forecast/analysis system, something like: . . . data is assimilated in 2 day analysis part of the run, which then is run freely as a 5 day forecast, with the end of the analysis used as initial conditions for the next forecast cycle. We use the analysis period to assess the impact of ssh, and then compare the errors in the 1st, 3rd and 5th day of the forecast period). . .

Page 7: section 2.3.2. Need to give more information on the current Nadir altimeters: how often do they pass over a particular place? How wide is the footprint? How wide is it between passes (compare to the newer ones)? What does 1Hz mean in terms of km's? You say the wide swath altimeters have 20.9 day repeats, and 7km – does this imply that the same is true for the nadir satellites? Perhaps a table giving these details might be useful.

Page 7: section 3. Did you consider an OSSE assimilating TS and SST, but not SSH? Assuming correcting the ocean temperature structure won't affect the SSH through expansion, but it may constrain eddies, which would affect the SSH?

Page 8: Line 26. Clarify exactly what variance you mean. I assume you mean a spatial map of temporal variance. Maybe add an equation along the lines of:

$$\frac{\sum_{t=0}^{t=T}(ssh_{OSSEk}(x,y,t)-ssh_{NR}(x,y,t))^2}{n_t} - \left(\frac{\sum_{t=0}^{t=T}(ssh_{OSSEk}(x,y,t)-ssh_{NR}(x,y,t))}{n_t}\right)^2$$

$$Var(NR) = \frac{\sum_{t=0}^{t=T}ssh_{NR}(x,y,t)^2}{n_t} - \left(\frac{\sum_{t=0}^{t=T}ssh_{NR}(x,y,t)}{n_t}\right)^2$$

Page 9: Line 31-32. It is good to see you talk about what the improved statistics mean physically (in terms of ocean features). You should do more of this in the paper.

Page 20: Figure 1. I suggest you use a perceptually uniform colour map, and don't use the jet/rainbow style colour map. This applied to all the map figures.

**3  Technical corrections**

Page 2: Line 6. "these" and "source" need to agree – perhaps "these unique sources".

Page 2: Line 28. Change "could not be suitable" to "may not be suitable"?

Page 3: Line 14. Remove the space before the semi colon "NEMO ; Madec" vs "NEMO;

Madec".

Page 3: Line 15. Perhaps change "'nature" run' to "'natural" run'?

Page 4: Line 4. Remove "a" from "In this study a particular attention.."

Page 4: line 25. You say the "last version". Do you really mean Last, latest or previous?

Page 4: line 26. Remove extra space, change "used , the" with "used, the"

Page 6, Line 7. Remove extra space after reference.

Page 6: line 29. Should it be daily average sst? I guess it depends on the assimilation system?

Page 8: line 4-5. Incorrect phrasing, please rephrase "As already mentioned, SWOT-like data have a temporal resolution which could not allow to resolve correctly the evolution of mesoscale structures"

Page 8: line 8.Is order of magnitude the correct term?

Page 8: Line 13-16. Justify the values of the improved radar interferometer error values in OSSE4. Is this a possible improvement?

Page 8: line 15. Change 'an halved' to 'a halved'

Page 8: line 19. Do you mean NR? If so, change for consistency.

Page 8: line 20. Perhaps helpful to say steep bathymetric slope or deep sea-bed slope – might otherwise confuse some readers.

Page 8: line 21. Change word order to 'was also captured'

Page 8: Line 25. Perhaps call var* relative variance?

Page 9: Line 8. Change 'an higher' to 'a higher'

Page 9: Line 17. Is 20-30

Page 9: Line 26. Is the 6

Page 10: Line 3. Add a reference to Table 4.

Page 10: Line 10. Is this the correct table?

Page 10: Line 15. Is 19W too close to your lateral open boundary?

Page 10: Line 21-23. The wording is confusing in this sentence, feels like it's the wrong way around.

Page 10: Line 21. Change word order 'to evaluate also' to 'to also evaluate'.

Page 10: Line 24. Looks like you can see the impact to 50km, although to a much smaller level. Perhaps add a qualifier here, or weaken.

Page 10: Line 27. Also add a reference to table 7 – something like "(Figure 7, left panel; Table 7)".

Page 10: Line 30-32. What about the difference between OSSE2 and OSSE3? I assume you mean it's interesting to notice that the difference is small, but maybe no... clarify.

Page 11: Line 17. Change word order "increased significantly" to "significantly increased".

Page 12: Line 2. We've just been talking about Table 5, do you need this last sentence?

Page 13: Line 5. Do you mean OSSE4 when you say "accurate" if so, clarify. Also applied to Line 27

Page 13: Line 22. Change word order from "observed also" to "also observed".

**4 Figures**

Page 21: Line 3. Use left, centre, right to describe the upper panels. It is unclear using semi-colons.

Page 22: Figure 4. Given the whitespace in the upper left, perhaps add OSSE1 in blue. . . OSSE4 in green text. This applies to figure 5 and 7. It's good you've used the same colour ordering for these panels. Check the colour for colour blindness. Perhaps removed the 100's of x ticks.

Page 22: Line 3 (caption for Figure 4). Change "(blue lines)" to "(blue line)".

Page 23: Line 2. Type – Frebruary – change to February

Page 25: Figure 7. Break Y axis on the left hand panel, (i.e. y values of 0-4 and then 4-10) to capture the top of the grey line.

Page 25: Line 2-3. Confusing text. . . perhaps say something like ". . . February-December 2009. The results for experiments OSSE0. . . OSSE4 (green lines) are shown at the spectral window between 400km and 12km."

**5 Tables**

Page 29. Table 5. Add km to the last 3 columns (280km, 155km, 125km. . .)

Page 29: Table 6. Add columns for current magnitude.

---

## Referee Comment (RC2) · Anonymous Referee #2 · 21 Aug 2018

This paper evaluates the impact of assimilating wide-swath altimetry to improve ocean analysis and prediction. Observing System Simulation Experiment (OSSE) methodology is used in a regional setting in the northeastern Atlantic Ocean to evaluate the impact of this future observing technology in comparison to the impact of existing along-track altimetry. Errors in ocean analyses are further reduced by up to 50% over the reduction achieved by assimilating the existing constellation of along-track altimeters. Substantial error reduction is maintained by short-term ocean forecasts initialized by these data-assimilative ocean analyses.

This is a significant paper for two reasons. First, the authors followed rigorous procedures with respect to the design and validation of the OSSE system to ensure that credible impact assessments are obtained. Second, wide-swath altimetry is an impor-

tant new technology that holds the promise of significantly improving the analysis and prediction of ocean mesoscale variability. The experimental design is reasonable. This paper provides an important early quantitative assessment of the expected improvement when wide-swath altimetry becomes operational. The paper is clearly written and I have no significant editorial recommendations.

For these reasons, I recommend publication as is.

―――――――――――――――

---

## Author Comment (AC1) · 18 Oct 2018

*General comments*

*There are currently 3 nadir altimeters in orbit, which provide along track measurements of Sea-Surface Height (SSH). However, due to their narrow band nature, and the repeat times, the gaps between tracks are large compared to the ocean mesoscale – thus only limited spatial and temporal coverage is available. Wide swath altimeters are being developed to which would provide much greater SSH data.*

*This paper looks at the implication of such data being available to constrain model solutions for the IBI region. They note that similar studies are needed for the global scale, and for other regions.*

*They run a series of Observations Synthetic Simulation Experiments to assess the impact of assimilating the current constellation of altimeters, and future wide band altimeters. I found the experimental design satisfying, and analysis convincing.*
We thank the Reviewer for the positive comments. Here follows a point-by-point list of response for each specific comment *(highlighted in bold+italic)*, as well as a marked-up manuscript version with tracked changes.

*Specific comments*

*I found the paper well thought out, clear and useful, and would recommend it for publication with corrections. I think it had scientific significance, scientific quality.*
Thanks.

*The presentation quality is also quite good, although I think it needs to be proof read by another native-English speaker, as there are sections that are incorrect or unclear.*
We thank the Reviewer for this suggestion. A native-English speaker (professional) proofreader revised the manuscript. The text has changed in several parts. All the changes can be noticed in the marked-up manuscript version with tracked changes.

*I think they need to reproduce the graphics with a better colour-map – jet/rainbow is very misleading.*
We thank the Reviewer for this comment. The maps are shown now with a perceptually proportional palette.

*I thought there should be a bit more description of the altimeters and of the ocean forecasting/analysis/reanalysis system.*
We thank the reviewer for this comment. A table (Table 2) was added, showing the characteristics of the altimetry missions considered in this study and the text has changed to describe more the ocean analysis and forecasting system considered in this study (Page 3: lines 31-35).

*I think there could be a little more discussion of what the statistics mean physically, in terms of location, mechanism etc.*

We thank the Reviewer for the appropriate comment. The text has changed discussing more about the physical meaning of the impacts observed (Page 9: lines 25-27; Page 10: lines 6-9; Page 10: lines 24-25).

*I have outlined these specific comments here.*

*Page 4: End of introduction. Later in the paper when you start talking about forecast errors, I realised that it was not a reanalysis study – I wonder if this should be further clarified in the introduction . . . perhaps a sentence of two describing the forecast/analysis system, something like: . . . data is assimilated in 2 day analysis part of the run, which then is run freely as a 5 day forecast, with the end of the analysis used as initial conditions for the next forecast cycle. We use the analysis period to assess the impact of ssh, and then compare the errors in the 1st, 3rd and 5th day of the forecast period). . .*

We thank the Reviewer for the appropriate comment. The text in the Introduction has changed (Page 3: lines 31-35):

"The system used in this study differs from operational systems, in the sense that the same atmospheric forcings are used both in the hindcast and forecast. First, the system is run freely as a 5-day forecast to compute innovations (difference between observations and model background). Then, data is assimilated in a 5-day analysis, with the end of the analysis used as initial conditions for the next forecast cycle. Both analyses and forecastings fields are stored to assess the impact of altimetry data in ocean analyses and forecasting skill (no data-assimilation, same atmospheric forcings) in the 1st, 3rd and 5th day of the forecast period."

*Page 7: section 2.3.2. Need to give more information on the current Nadir altimeters: how often do they pass over a particular place? How wide is the footprint? How wide is it between passes (compare to the newer ones)? What does 1Hz mean in terms of km's? You say the wide swath altimeters have 20.9 day repeats, and 7km – does this imply that the same is true for the nadir satellites? Perhaps a table giving these details might be useful.*

We thank the Reviewer for this comment. We added a Table describing the characteristics of the satellite altimetry missions considered (Table 2). The text has also changed in the manuscript:

- page 7: lines 7-9. "Conventional altimetry data were derived from sampling the NR over the theoretical tracks of the satellite missions Jason 2, Cryosat 2 and Sentinel 3a, with a sampling frequency of 1 Hz (∼7 km spatial sampling; e.g. Roblou et al. 2011).".

- page 7: lines 15-16: "The specifics for each satellite altimetry mission considered in this study are detailed in Table 2."

*Page 7: section 3. Did you consider an OSSE assimilating TS and SST, but not SSH? Assuming correcting the ocean temperature structure won't affect the SSH through expansion, but it may constrain eddies, which would affect the SSH?*

We thank the Reviewer for this comment. The Reviewer is right, ocean temperature structure can constrain eddies and as a consequence can affect the SSH. On the other hand, in literature it is known that satellite altimetry observations has a major impact, with respect to the other observations systems,

to constrain the ocean variability due to mesoscale structures (Lea et al., 2014, Oke et al., 2015, Verrier et al., 2017) and nowadays they are assimilated in realistic ocean analyses and forecasting systems (Le Traon et al., 2015). The OSSEs performed in this study were designed in order to be representative of the evolution of errors found in real ocean analysis and forecasting systems. The system used here (IBI) is adopted by CMEMS to provide operational analyses, forecasts and ocean reanalysis where all the best quality flagged observations (e.g. SSH, TS, SST) are ingested by system. In order to perform realistic OSSEs, we decided on purpose to assimilate TS, SST and SSH in all the OSSEs, and to quantify the impact of wide-swath altimeters looking at the variations of the error due to the different SSH synthetic observations considered. We adopted this approach to achieve the objective of the study, which was to asses the impact of future constellations with respect to the current altimeters. For these reasons in this study we didn't consider an OSSE assimilating only TS and SST.

We thank the Reviewer for this comment. The text has changed and further discussion about the physical meaning of the improved statistics is given at:

- Page 9: lines 25-27. "Large errors were observed in occurrence of the main features of the ocean circulation in the IBI region, both in the northern (e.g. North Atlantic Drift) and southern part of the domain (e. g. Azores Current), as well as in the Bay of Biscay."

- Page 10: lines 6-9. The impact of wide-swath altimetry measurements can be noticed over the entire spatial domain and in particular in the areas of the ocean characterized by the signature of the North Atlantic, Azores and Canary Currents and in the occurrence of mesoscale eddies in the Bay of Biscay, where large errors were observed considering only nadir altimeters."

- Page 10: lines 24-25. "This involve a further improvement of the representation of the ocean dynamics due to the main ocean currents and mesoscale structures which characterize the SSH variability in the IBI region."

*Page 20: Figure 1. I suggest you use a perceptually uniform colour map, and don't use the jet/rainbow style colour map. This applied to all the map figures.*
Thanks for this appropriate comment. Map figures are shown using a perceptually uniform colour map, except those showing the synthetic observations (Figure 1 and 2) to differ from map figures which show the results of this study (Figure 3, 6 and 7).

*Technical corrections*

*Page 2: Line 6. "these" and "source" need to agree – perhaps "these unique sources".*
Corrected.

*Page 2: Line 28. Change "could not be suitable" to "may not be suitable"?*
Corrected.

*Page 3: Line 14. Remove the space before the semi colon "NEMO ; Madec" vs "NEMO;Madec".*
Corrected.

*Page 3: Line 15. Perhaps change "'nature" run' to "'natural" run'?*
Corrected.

*Page 4: Line 4. Remove "a" from "In this study a particular attention.."*
Corrected (Page 4: line 8).

*Page 4: line 25. You say the "last version". Do you really mean Last, latest or previous?*
Thank for this comment. We mean the "latest": the text has changed, referring to the "latest version" (Page 4: line 29).

*Page 4: line 26. Remove extra space, change "used , the" with "used, the"*
Corrected (Page 4: line 30) .

*Page 6, Line 7. Remove extra space after reference.*
Corrected (Page 6: line 11).

***Page 6: line 29. Should it be daily average sst? I guess it depends on the assimilation system?***
Thanks for this comment. It depends on the assimilation system used. The text has changed (Page 6 : lines 31-32; Page 7: line 1):
"To accurately assess the impact of satellite altimetry data on ocean analyses and forecasts, the same synthetic observations of SST, T and S profiles were considered in all the experiments. An SST map representing a daily average is assimilated during each 5-day assimilation cycle.".

***Page 8: line 4-5. Incorrect phrasing, please rephrase "As already mentioned, SWOT-like data have a temporal resolution which could not allow to resolve correctly the evolution of mesoscale structures"***
Thanks for this comment. The text has changed (Page 8: lines 9-11):
"As already mentioned, SWOT-like data has a temporal resolution which does not allow the evolution of mesoscale structures to be resolved correctly".

***Page 8: line 8. Is order of magnitude the correct term?***
Thanks. The expression "order of magnitude" was omitted (Page 8: line 14).

***Page 8: Line 13-16. Justify the values of the improved radar interferometer error values in OSSE4. Is this a possible improvement?***
Thanks for this comment. The error values in OSSE4 represent a possible scenario analyzed by Thales Alenia Space (TAS) to develop European wide-swath altimetry concepts, with less stringent noise requirements compared to SWOT mission. The text has changed (Page 8: lines 18-23):
"In order to investigate the sensitivity of the ocean analysis and forecasting system to the error of a wide-swath altimetry instrument, a dedicated OSSE, hereafter OSSE4, was performed considering a satellite constellation as in OSSE3 but assuming a radar interferometer error of one half the order of magnitude (0.4 - 1 cm) with respect to the other OSSEs (2 x KaRIN error). The error values in OSSE4 represent one of the solutions analyzed by TAS, as part of this ESA study, to develop European wide-swath altimetry concepts. The experimental set-up used in this study is detailed in Table 2."

***Page 8: line 15. Change 'an halved' to 'a halved'***
Corrected (previous answer).

***Page 8: line 19. Do you mean NR? If so, change for consistency.***
Corrected (Page 8: line 26).

***Page 8: line 20. Perhaps helpful to say steep bathymetric slope or deep sea-bed slope might otherwise* confuse some readers.***
Thanks for the comment. The text has changed (Page 8: lines 26-27): "...steep bathymetric slope separating...".

***Page 8: line 21. Change word order to 'was also captured'***
Corrected (Page 8: lines 28-29).

***Page 8: Line 25. Perhaps call var* relative variance?***
Thanks. The text has changed (Page 9: line 2): "...we considered the relative variance VAR* defined...".

***Page 9: Line 8. Change 'an higher' to 'a higher'***
Corrected (Page 9: line 19).

***Page 9: Line 17. Is 20-30***
Corrected (Page 10: Line 1).

***Page 9: Line 26. Is the 6***
Thanks. The text has changed (Page 10: line 16): "V AR∗ was also 6% lower for OSSE2 than for OSSE1."

***Page 10: Line 3. Add a reference to Table 4.***
Corrected referring to Table 5 (Page 10: line 26).

***Page 10: Line 10. Is this the correct table?***
*Thanks for the comment. The reference to Table was correct, but the text was misleading. The text has changed (Page 10: line 34):"The results of the impact of wide-swath altimetry data on the SSH in ocean analysis are summarized in Table 4.".*

***Page 10: Line 15. Is 19W too close to your lateral open boundary?***
The reviewer is right, 19W can be considered close to the lateral open boundary. The sub-domain was selected in order to consider an open ocean region characterized by high mesoscale acitivity in the IBI domain. This was a compromise between the extension of the area considered (which affect the spatial scales in a spectral analysis), and the distance from the lateral boundary.

***Page 10: Line 21-23. The wording is confusing in this sentence, feels like it's the wrong way around.***
Thanks for this comment. The text has changed (Page 11: lines 11-13): "Here the reduction of the error at the different wavelength (ERspec) is defined as the percentage decrease of the error with respect to OSSE0 (Table 6), in order to asses also the impact of nadir altimeters".

***Page 10: Line 21. Change word order 'to evaluate also' to 'to also evaluate'.***
Corrected (previous answer).

***Page 10: Line 24. Looks like you can see the impact to 50km, although to a much smaller level. Perhaps add a qualifier here, or weaken.***
Thanks for this comment. The text has changed (Page 11: lines 14-15): "Considering nadir altimeters (blue curve), the impact on ocean analysis is noticeable at spatial scales down to 100 km, while is weaken at wavelengths of 50 km."

***Page 10: Line 27. Also add a reference to table 7 – something like "(Figure 7, left panel; Table 7)".***
Corrected (Page 11: line 17).

***Page 10: Line 30-32. What about the difference between OSSE2 and OSSE3? I assume you mean it's interesting to notice that the difference is small, but maybe no. . .clarify.***
Thanks for this appropriate comment. The text has changed (Page 11: lines 20-23): "Here it is also interesting to note that the difference between OSSE2 and OSSE3 (purple and red lines) is small, while is significant comparing OSSE2 and OSSE4 (green line) showing the impact of the higher repeat cycle of the SSH measurements and the sensitivity of system to the error of a wide-swath instrument.

***Page 11: Line 17. Change word order "increased significantly" to "significantly in-creased".***
Corrected (Page 12: line 7).

***Page 12: Line 2. We've just been talking about Table 5, do you need this last sentence?***
Thanks for this comment. The sentence was omitted.

***Page 13: Line 5. Do you mean OSSE4 when you say "accurate" if so, clarify. Also applied to Line 27***
Thanks for this comment. The text has changed (Page 13: line 26; Page 14: lines 2-3 and 17) referring explicitly to OSSE4.

***Page 13: Line 22. Change word order from "observed also" to "also observed".***
Corrected (Page 14: line 11).

**Figures**

***Page 21: Line 3. Use left, centre, right to describe the upper panels. It is unclear using semi-colons.***
Corrected. The text in the caption of Figure 2 has changed: "Top panels: satellite altimetry spatial coverage during one assimilation cycle (5 days); left: Jason2, Cryosat 2 and Sentinel 3a; central: Swath-1; right: Swath-1 & Swath-2."

***Page 22: Figure 4. Given the whitespace in the upper left, perhaps add OSSE1 in blue. . . OSSE4 in green text. This applies to figure 5 and 7. It's good you've used the same colour ordering for these panels. Check the colour for colour blindness. Perhaps removed the 100's of x ticks.***
Thanks. Figure 4, 5 and 7 have changed showing the experiments legend. In theFigures, the legend has changed to take into account color blindness (e.g. https://rdrr.io/cran/ggthemes/man/colorblind.html):
- OSSE1 (blue; RGB: #56B4E9) ;
- OSSE2 (orange; RGB: #E69F00) ;
- OSSE3 (purple; RGB: #CC79A7) ;
- OSSE4 (green; RGB: #009E73).
The 100's of x ticks were removed in Figure 4.

***Page 22: Line 3 (caption for Figure 4). Change "(blue lines)" to "(blue line)".***
Corrected.

***Page 23: Line 2. Type – Frebruary – change to February***
Corrected (caption for Figure 5).

***Page 25: Figure 7. Break Y axis on the left hand panel, (i.e. y values of 0-4 and then 4-10) to capture the top of the grey line.***
Thanks. Left panel in Figure 7 has changed accordingly.

***Page 25: Line 2-3. Confusing text. . . perhaps say something like ". . . February-December 2009. The results for experiments OSSE0. . . OSSE4 (green lines) are shown at the spectral window between 400km and 12km."***
Thanks for this comment. The caption of Figure 7 has changed accordingly.

*Tables*

***Page 29. Table 5. Add km to the last 3 columns (280km, 155km, 125km. . .)***
Corrected (Table 6).

***Page 29: Table 6. Add columns for current magnitude.***
Corrected (Columns 1-3 and in Table 7).

[revised manuscript text omitted]

---

## Author Comment (AC2) · 18 Oct 2018

*This paper evaluates the impact of assimilating wide-swath altimetry to improve ocean analysis and prediction. Observing System Simulation Experiment (OSSE) methodology is used in a regional setting in the northeastern Atlantic Ocean to evaluate the impact of this future observing technology in comparison to the impact of existing along-track altimetry. Errors in ocean analyses are further reduced by up to 50% over the reduction achieved by assimilating the existing constellation of along-track altimeters. Substantial error reduction is maintained by short-term ocean forecasts initialized by these data-assimilative ocean analyses.*

*This is a significant paper for two reasons. First, the authors followed rigorous procedures with respect to the design and validation of the OSSE system to ensure that credible impact assessments are obtained. Second, wide-swath altimetry is an important new technology that holds the promise of significantly improving the analysis and prediction of ocean mesoscale variability. The experimental design is reasonable. This paper provides an important early quantitative assessment of the expected improvement when wide-swath altimetry becomes operational.*

*The paper is clearly written and I have no significant editorial recommendations.*

*For these reasons, I recommend publication as is.*

We thank the Reviewer for the positive comments *(highlighted in bold+italic)* and for considering this paper as a significant study.

---

## Author Response (AR2)

**Author's response**

*Topic Editor Decision: Publish subject to technical corrections* (26 Oct 2018) by John M. Huthnance

*Comments to the Author:*

*Thank-you for your revised manuscript attending to all the review comments. Now I am just asking you to consider "Technical Corrections (see below) after which it should go directly into the publication procedure (i.e. no more intervention by me). There will be copy-editing and you should check that your intended meaning is retained. Thank-you for publishing in Ocean Science.*
*Yours sincerely*
*John Huthnance*
We thank the Editor for the appropriate "Technical Corrections" (highlighted in **bold+italic**). Here follows a point-by-point list of response, as well as a marked-up manuscript version with tracked changes.

Thank-you for considering this paper suitable for publication in Ocean Science.
Best regards,
Antonio Bonaduce (on behalf of all co-authors).

*Page 1*
*Line 4. "simulation" (delete final "s")*
Corrected.
*Line 14. Better ". . 5-day forecasts, compared with a single . ."?*
Corrected.
*Line 16. Delete 2nd "to"*
Corrected.
*Line 18. Better ". . propagated down the water column . ."*
Corrected.

*Page 2*
*Line 6. Omit "," after "(Bell et al., 2015)"*
Corrected.
*Lines 21-22. Better ". . resolution, to wavelengths as short as 20 km . ."?*
Corrected.
*Line 23. "involve" -> "make" ? (usually "make measurements" in English)*
Corrected.
*Line 30. "dynamic" -> "dynamical"?*
Corrected.

*Page 6 line 23. "background-error" ?*
Corrected.

*Page 7*
*Line 18. ". . were obtained by sampling . ."*
Corrected.
*Page 8 lines 2-3. "Better ". . data assimilation in . ." ?*

Corrected.

*Page 10*
*Lines 11-12. Better ". . with depth greater than 200 metres . ."*
Corrected.
*Line 12. From (2) I do not think that OSSE1 ER\* is meaningful so cannot be compared with. End the sentence at "~ 35%."?*
Corrected.
*Line 25. Better "significant ER\* (~ 28 %) in the ocean forecast till the 5th day of forecast, . ."*
Corrected.
*Line 30. Better "This involve" -> "OSSE4 gave"?*
Corrected.

*Page 11*
*Line 15. Not "while is weaken". Maybe "but the impact is weaker"*
Corrected.
*Line 21. Not "while is significant". Maybe ". . small, but the difference between OSSE2 and OSSE4 (green line) is significant, showing the impact"*
Corrected.
*Line 26. "lowest" -> "least"*
Corrected.
*Line 28. Better ". . reduction of error (ERspec ~ 30 %). On the other hand . ." ?*
Corrected.
*Line 29. Better ". . to better resolve ocean dynamics . ." and add "," after "variability"*
Corrected.

*Page 12*
*Line 7. ". . data significantly increased the coherence . ."*
Corrected.
*Line 31. I prefer ". . (OSSE1); large errors were observed . ."*
Corrected.

*Page 13*
*Lines 4, 6. Better to add "ER\*" before the values "~28%" , "~35%".*
Corrected.
*Line 7. Omit ","*
Corrected.
*Line 17. "Cryosat2" ?*
Corrected.
*Line 24. "concerns"*
Corrected.
*Line 30. "constellation of . . . and one wide-swath mission (3N+1S) for the first day . ." (order)*
Corrected.

*Page 14*
*Line 16. "reduction of the error down to ~ 10% was observed". This means the remaining error was only 10% of some reference value (which should be stated). But I think you might mean the reduction was 10%, i.e. "reduction of the error by ~ 10% was observed".*
Corrected.

*Line 17. "with respect to" -> "compared with"*
Corrected.
*Line 19. "interferometer" (spelling)*
Corrected.

*Figure 2 caption*
*Line 2. ". . right: Swath-1+ Swath-2. Bottom . ." ?*
Corrected.
*Lines 2, 4, 5. Better "across-swath" (twice) and "along-swath".*
Corrected.

*Figure 8 caption line 3. OSSE4 is at bottom but not "right"..*
Corrected.

*Table 2. The units "days", "km", "km" would be better alongside the respective headings "RC", "FP", "CTS".*
Corrected.

*Table 3 caption line 3. Delete duplicate "the".*
Corrected.
*Table 3. In row OSSE3 column T&S, should be "YES"?*
Corrected.

*Table 4 caption line 3. ". . in each experiment . ."*
Corrected.

[revised manuscript text omitted]